# N-Acetylcysteine Reduces the Pro-Oxidant and Inflammatory Responses during Pancreatitis and Pancreas Tumorigenesis

**DOI:** 10.3390/antiox10071107

**Published:** 2021-07-11

**Authors:** Marie-Albane Minati, Maxime Libert, Hajar Dahou, Patrick Jacquemin, Mohamad Assi

**Affiliations:** Liver and Pancreas Differentiation (LPAD) Group, de Duve Institute, Université Catholique de Louvain, 1200 Brussels, Belgium; marie-albane.minati@uclouvain.be (M.-A.M.); maxime.l.libert@uclouvain.be (M.L.); hajar.dahou@uclouvain.be (H.D.)

**Keywords:** antioxidants, cancer, edema, inflammation, pancreas

## Abstract

Pancreatitis, an inflammation of the pancreas, appears to be a main driver of pancreatic cancer when combined with *Kras* mutations. In this context, the exact redox mechanisms are not clearly elucidated. Herein, we treated mice expressing a *Kras^G12D^* mutation in pancreatic acinar cells with cerulein to induce acute pancreatitis. In the presence of *Kras^G12D^*, pancreatitis triggered significantly greater redox unbalance and oxidative damages compared to control mice expressing *wild-type* *Kras* alleles. Further analyses identified the disruption in glutathione metabolism as the main redox event occurring during pancreatitis. Compared to the *wild-type* background, *Kras^G12D^*-bearing mice showed a greater responsiveness to treatment with a thiol-containing compound, N-acetylcysteine (NAC). Notably, NAC treatment increased the pancreatic glutathione pool, reduced systemic markers related to pancreatic and liver damages, limited the extent of pancreatic edema and fibrosis as well as reduced systemic and pancreatic oxidative damages. The protective effects of NAC were, at least, partly due to a decrease in the production of tumor necrosis factor-α (TNF-α) by acinar cells, which was concomitant with the inhibition of NF-κB(p65) nuclear translocation. Our data provide a rationale to use thiol-containing compounds as an adjuvant therapy to alleviate the severity of inflammation during pancreatitis and pancreatic tumorigenesis.

## 1. Introduction

Acute pancreatitis is defined as a transient and local inflammation of the pancreas characterized by immune cell infiltration and edema [1]. In the absence of appropriate medical care, acute pancreatitis can lead to systemic multiorgan failure and death. The mortality of patients with severe pancreatitis is around 5% [2]. Pancreatitis first affects acinar cells; it causes acinar-to-ductal metaplasia (ADM) and promotes an excessive release of digestive acinar enzymes, which results more widely in pancreatic damage and dysfunction [3]. In the presence of oncogenic mutations in the *Kirsten rat sarcoma* (*Kras*) gene, pancreatitis has been shown to promote the development of pancreatic intraepithelial neoplasia (PanIN) and pancreatic ductal adenocarcinoma (PDAC) from acinar cells [4,5,6]. 

Emerging studies confirm the importance of reactive oxygen species (ROS) in PDAC pathophysiology [7,8]. Yet, the precise redox changes driven by pancreatitis during PDAC initiation are still not clearly elucidated. Indeed, multiple studies have characterized the redox alterations occurring during inflammation-induced metabolic dysfunction and pancreatitis in *wild-type* rodents [9,10,11,12,13,14,15,16]; however, there is a lack of information regarding the main redox changes taking place during the initiation of pancreas tumorigenesis (presence of *Kras* mutations). To fill this gap of knowledge in the field, we herein compared mice expressing *Kras wild-type* alleles (control mice) with those expressing acinar-specific mutated *Kras^G12D^* oncogenic allele. We found that pancreatitis induced greater redox unbalance and oxidative damages during pancreas tumorigenesis in mice bearing *Kras^G12D^* mutation. Strikingly, *Kras^G12D^* mice were more responsive to supplementation with the thiol-rich compound, N-acetylcysteine (NAC), as evidenced by a reduction in inflammation severity and systemic and pancreatic oxidative damages. Therefore, using this comparative approach, our study demonstrates that *Kras^G12D^* mutation aggravates redox unbalance in the presence of pancreatitis. It also provides a rationale to use thiol-containing compounds as an adjuvant therapy, not only for pancreatitis, but also in the context of pancreatic tumorigenesis.

## 2. Materials and Methods

### 2.1. Human Material and Ex Vivo Culture

Pancreata from human heart-beating cadaveric donors were processed by the Beta Cell Bank (Brussels, Belgium) affiliated to the Euro-Transplant Foundation (Leiden, The Netherlands). A full written consent was obtained and the medical ethical committee of the Vrije Universiteit Brussel Hospital approved the use of donor pancreata (ethic approval number: B.U.N. 143201732606). Culture protocol was performed as previously described [17]. This culture spontaneously induces ADM and mimics the molecular mechanisms driven by inflammation. Cells were maintained in 3D suspension culture for 3 days in Advanced RPMI 1640 medium (12633-012, Thermo Fisher, Merelbeke, Belgium) supplemented with 5% fetal bovine serum (FBS) (F7524, Sigma Aldrich, Overijse, Belgium), 1% penicillin-streptomycin (15070-063, Life Technologies, Merelbeke, Belgium) and 0.1 mg/mL soybean trypsin inhibitor (17075-029, Life Technologies, Merelbeke, Belgium), at 37 °C and 5% CO_2_. On the day of isolation (day 0), 100 µL of cells, corresponding to the normal fraction, was snap-frozen in liquid nitrogen and stored at −80 °C, for future protein extraction.

### 2.2. Mice Models and Treatments

All procedures described were approved by the animal welfare committee of the University of Louvain Medical School (ethic number: 2017/UCL/MD/020). *Wild-type* (*WT*) and *ElastaseCreER-LSLKras^G12D^* (*Ela-Kras^G12D^*) mice were maintained in a CD1-enriched background; this mouse model allows the expression of a mutated *Kras^G12D^* allele specifically in acinar cells to study pancreas tumorigenesis process. *ElaCreER* and *LSLKras^G12D^* mice have been described previously [5,18]; these mice were bred in our animal facility to obtain an *Ela-Kras^G12D^* background. In select experiments, *ElaCreER* and/or *LSLKras^G12D^* mice were bred with LSLRosa^yellow fluorescence protein (YFP)/YFP^ mice to obtain *ElaCreER^YFP/YFP^ (Ela^YFP/YFP^)* and *ElaCreER-LSLKras^G12D; YFP/YFP^* (*Ela-Kras^G12D; YFP/YFP^*) mice; since the Elastase promoter is selectively expressed in pancreatic acinar cells, these mice allow the specific isolation and study of YFP-positive acinar cells. Adult *Ela^YFP/YFP^, Ela-Kras^G12D; YFP/YFP^* and *Ela-Kras^G12D^* mice (6-week-old) were first treated with tamoxifen (30 mg/mL in corn oil) by oral gavage (T5648-1G, Sigma Aldrich, Overijse, Belgium) and 4-hydroxytamoxifen (0.3 mg/mL in corn oil) by subcutaneous injection (H7904-25mg, Sigma Aldrich, Overijse, Belgium) to recombine the LSL stop cassettes and allow the expression of *Kras^G12D^* and/or *YFP* from their respective endogenous locus. To induce acute pancreatitis, *WT* and *Ela-Kras^G12D^* mice received intraperitoneal injections of cerulein (100-150 µL in phosphate buffered saline, PBS) (125 µg/kg; 24252, Eurogentec, Seraing, Belgium) for 5 days. We must note that lower amounts of cerulein could be injected depending on the used mouse strain (e.g., Black-6 mice could be injected with 60 µg/kg of cerulein). In addition to cerulein injections, a set of *WT* and *Ela-Kras^G12D^* mice received intraperitoneal N-acetylcysteine (NAC) (1 g/kg; A7250-50G, Sigma Aldrich, Overijse, Belgium) injections. For transcriptomic analyses on isolated YFP-positive acinar cells, *Ela^YFP/YFP^* and *Ela-Kras^G12D; YFP/YFP^* mice received cerulein injections for 3 days (125 µg/kg). Mice were sacrificed at 8 weeks of age by cervical dislocation and tissues were collected for subsequent analyses. 

### 2.3. Ex Vivo Culture of Primary Mouse Acinar Cells

Cultures of dissociated mouse pancreata that mimic ADM were isolated as previously described [19]. This culture spontaneously mimics the molecular mechanism driven by inflammation. Briefly, pancreata were cut into small pieces and digested with collagenase P (11213865001, Sigma Aldrich, Overijse, Belgium) (0.35 mg/mL of collagenase in Hank’s balanced salt solution (HBSS) buffer, 14025-100, Life Technologies, Merelbeke, Belgium). Collagenase incubation was performed for 15 min at 37 °C with 180 rotations per minute (rpm). The cell suspension was washed three times with HBSS/5% FBS and filtered through 500 µm (43-50500-50, ImTec Diagnostics, Antwerp, Belgium) and 100 µm (CLS431752-50EA, Sigma Aldrich, Overijse, Belgium) strain filters. Finally, cells were dropped gently on a cushion of 30% FBS and centrifuged at 1000 rpm, for 2 min. The day of isolation (day 0) is equivalent to normal pancreas. Metaplasia was induced by maintaining acinar cells in 3D suspension culture, for 3 days, in Advanced RPMI supplemented with 5% FBS, 1% penicillin-streptomycin and 0.1 mg/mL soybean trypsin inhibitor, at 37 °C and 5% CO_2_. 

### 2.4. Measurement of Reactive Oxygen Species (ROS) in Primary Acinar Cells 

Primary mouse acinar cells were isolated as described above and maintained in culture for 1 or 3 days. At the different timepoints, cells were washed with PBS and incubated for 30 min with 5 µM DCFDA probe at 37 °C (ab113851, Abcam, Cambridge, UK). Then, cells were washed once with PBS and pelleted by centrifugation (1200 rpm, 1 min). The pellet was resuspended in 1 mL PBS and cells were distributed in the wells of an opaque 96-well plate. Finally, DCFDA fluorescence was measured using a microplate reader (GloMax, Promega, Leiden, The Netherlands) at Excitation/Emission = 485/535 nm.

### 2.5. Detection of 4-Hydroxynonenal (4-HNE) Adducts by Immunoblot

This analysis was performed as described previously [20,21]. Proteins (50 µg) were separated on 12.5% SDS polyacrylamide gels. After transfer, PVDF membranes (ISEQ00010, Thermo Fisher Scientific, Merelbeke, Belgium) were incubated with a reducing solution (100 mM MOPS (M1254-100G, Sigma Aldrich, Overijse, Belgium) and 250 mM sodium borohydride, pH 8 (452882-100G, Sigma Aldrich, Overijse, Belgium)) for 15 min, at room temperature (RT). Membranes were then washed with ultrapure water and PBS (3 × 5 min). Membranes were incubated overnight at 4 °C with primary antibody (ab46545, Abcam, Cambridge, UK) in 5% low-fat milk diluted in PBS/0.25% Tween-20(P2287-500mL, Sigma Aldrich, Overijse, Belgium). The next day, membranes were incubated with the secondary antibodies for 1 h at RT in 5% low-fat milk diluted in PBS/0.25% Tween-20. 

### 2.6. Detection of Protein Carbonyls by Immunoblot

Protein carbonyls analysis was performed as described previously [21]. Briefly, 25 µg of total protein was derivatized with 1% Dinitrophenylhydrazine following manufacturer’s instructions (ab178020, Abcam, Cambridge, UK) and then separated by electrophoresis on 12.5% SDS polyacrylamide gels. Proteins were transferred on PVDF membranes, which were incubated with a blocking solution of 1% BSA in PBS/0.05% Tween-20, containing primary antibodies, overnight at 4 °C. The next day, membranes were washed and incubated with secondary antibodies in 5% low-fat milk diluted in PBS/0.05% Tween-20, for 1 h at RT.

### 2.7. Detection of Protein Expression by Immunoblot

Tissues and cells were lysed in 50 mM Tris-base-HCl, 150 mM NaCl and 1% NP40 (I8896-50ML, Sigma Aldrich, Overijse, Belgium) buffer. Protease and phosphatase (11836153001 and 04906837001, Sigma Aldrich, Overijse, Belgium) inhibitors were added just before lysis. Samples were maintained on ice during the procedure. Cell debris were pelleted by centrifugation (14,000× *g*, 10 min, at 4 °C). Proteins were quantified using a Bradford assay (23200, Thermo Fisher Scientific, Merelbeke, Belgium). Samples containing 30 to 50 µg of total proteins were electrophoresed on 7.5% to 12.5% SDS polyacrylamide gels. PVDF membranes were blocked with a solution of 5% low-fat milk diluted in Tris-buffered saline (TBS)/0.1% Tween-20. Membranes were incubated overnight with primary antibodies, at 4 °C. The used antibodies are listed in Table 1. The next day, membranes were washed with TBS/0.1% Tween-20 and incubated with the secondary antibodies for 1 h, at RT. After incubation, membranes were washed again and revealed using chemiluminescence (kits 34577 and 34094, Thermo Fisher Scientific, Merelbeke, Belgium). Pictures were taken with a Fusion Solo S machine (Vilber, Collégien, France). Densitometric analysis was performed using the Image Studio Lite Ver 5.2 software (LI-COR Biosciences, Lincoln, NE, USA).

### 2.8. Measurement of Reduced and Disulfide Glutathione 

Fresh pancreata were homogenized and deproteinated in a solution of 5% 5-sulfosalicylic acid (SSA; S2130-100G, Sigma Aldrich, Overijse, Belgium). After centrifugation (8000× *g*, 10 min), supernatant was diluted with double-deionized water to obtain a 0.5% SSA concentration. Then, reduced glutathione (GSH) and disulfide glutathione (GSSG) were measured according to manufacturer’s instructions (38185-1KT, Sigma Aldrich, Overijse, Belgium). Absorbance developed proportionally to the content of GSH or GSSG and was measured at 405/415 nm using a microplate reader (GloMax, Promega, Leiden, The Netherlands).

### 2.9. Measurement of Myeloperoxidase Activity (MPO)

MPO activity was measured according to the manufacturer’s instructions (ab150136, Abcam, Cambridge, UK). Pancreata were homogenized in MPO assay buffer and supernatant was collected after centrifugation (13,000 rpm, 10 min, 4 °C). Then, supernatant was added to a clear-bottom 96-well plate in presence of the MPO substrate and incubated at 25 °C for 30 min. Reaction was then stopped and absorbance measured at 412 nm using a microplate reader (GloMax, Promega, Leiden, The Netherlands).

### 2.10. Determination of TNF-α, Amylase, Aspartate Transaminase (AST) and Alanine Transaminase (ALT) Levels

Cultures of dissociated mouse pancreata, which mimic pancreatitis-induced metaplasia, were isolated as mentioned above. After isolation, acinar cells were treated with PBS (vehicle) or NAC (5 mM) for 3 days from the onset of the culture. After treatment, culture medium was collected for TNF-α ELISA measurements according to manufacturer’s instructions (430907, Biolegend, San Diego, CA, USA) [22]. TNF-α levels were also measured by ELISA in pancreas lysates from cerulein-treated *Ela-Kras^G12D^* mice supplemented with PBS or NAC. Absorbance was proportional to the content of TNF-α and was measured at 450 nm using a microplate reader (GloMax, Promega, Leiden, The Netherlands). Amylase, AST and ALT levels were measured in the plasma of cerulein-treated *wild-type* and *Ela-Kras^G12D^* mice, supplemented with PBS or NAC, using DRI-CHEM machine (NX500i, Fujifilm, Tokyo, Japan). To obtain plasma, blood was collected using EDTA-coated material and centrifugated at 1500× *g* for 10 min at 4 °C. The supernatant corresponding to the plasma fraction was then transferred to a new tube and used for measurements. 

### 2.11. Fluorescence Activated Cell Sorting (FACS), RNA Extraction and Quantitative PCR

Pancreata of *Ela^YFP/YFP^* and *Ela-Kras^G12D; YFP/YFP^* mice were digested using collagenase P (11213865001, Merck Life Science, Overijse, Belgium) (0.6 mg/mL), as previously described [23]. Briefly, an EGTA-buffered solution was injected in the main duct for optimization of pancreas dissociation. Pancreata were digested in calcium (Ca^2+^) buffer for 15–20 min, at 37 °C. After washing, the cells were filtered through a 35-µm cell strainer and sorted by FACS (FACSAria^TM^ III cell sorter, BD Biosciences, Erembodegem, Belgium). RNA extraction was performed using a column-based protocol (AM1912, Thermo Fisher, Merelbeke, Belgium) on YFP-positive acinar cells sorted from *Ela^YFP/YFP^* and *Ela-Kras^G12D; YFP/YFP^* pancreata treated or not with cerulein for 3 days. Reverse transcription reaction was performed on 150–250 ng of total RNA using the M-MLV reverse transcriptase (M1705, Promega, Leiden, The Netherlands). Quantitative PCR was carried out in a final volume of 10 µL (1 µL neosynthetized cDNA, 2 µL primers 10 µM, 5 µL Sybergreen KAPA mix 2X (KK4601, Sigma Aldrich, Overijse, Belgium), and 2 µL nuclease-free water) using the CFX96 Real-Time System thermocycler (C1000, Biorad, Temse, Belgium). The expression of target genes was normalized to the *ribosomal protein L4* (*Rpl4*) and *glyceraldehyde 3-phosphate dehydrogenase* (*Gapdh*) housekeeping genes. Relative expression was calculated using the ΔΔCt method. Primers used in the study are listed in Table 2.

### 2.12. Bulk RNA-Sequencing and Bioinformatics

The RNA Integrity Number (RIN) of total extracted RNA was measured using Agilent 2100 Bio-analyzer (Agilent Technologies, Santa Clara, CA, USA). Samples with a RIN ≥ 7 were selected for RNA-SEQ experiments, which were performed by the Novogene Company. Sequencing libraries were generated using the rRNA-depleted RNA by NEBNext^®^UltraTM Directional RNA Library Prep Kit for Illumina^®^ (E7420L, New England Biolabs, Ipswich, MA, USA) following manufacturer’s recommendations. Briefly, fragmentation was carried out using divalent cations under elevated temperature in NEBNext First Strand Synthesis Reaction Buffer (E7525L, New England Biolabs, Ipswich, MA, USA). First strand cDNA was synthesized using random hexamer primer and M-MuLV Reverse Transcriptase (M0253L, New England Biolabs, Ipswich, MA, USA). Second strand cDNA synthesis was subsequently performed using DNA polymerase I and RNase H. In the reaction buffer, dNTP with dUTP replacing dTTP were used. Then, NEBNext adaptors (E7335L, New England Biolabs, Ipswich, MA, USA) with hairpin loop structure were ligated to the adenylated 3′ ends of DNA fragments. The cDNA fragments of preferentially 150~200 bp in length were purified using AMPure XP system (Beckman Coulter, Brea, CA, USA). Library preparations were sequenced on an Illumina Hiseq platform and paired-end reads were generated.

For data analysis, clean data (clean reads) were obtained by removing reads containing adapter, reads containing poly-N, and low-quality reads from the raw data. All the downstream analyses were based on clean data of high quality. Reference genome and gene model annotation files were directly downloaded from the genome website. Index of the reference genome was built using Bowtie v2.0.6 [24,25] and paired-end clean reads were aligned to the reference genome using TopHat v2.0.9 [26]. Then, Cuffdiff (v2.1.1) was used to calculate fragments per kilo-base of exon per million fragments mapped (FPKMs) of coding genes in each sample [27]. KOBAS software was used to test the statistical enrichment of differentially expressed genes in KEGG pathways [28]. RNA-SEQ data can be downloaded using the following accession numbers: GSE163254 and GSE163263 [29]. 

### 2.13. Histology and Immunohistochemistry

Dissected pancreata were fixed in 4% paraformaldehyde (PFA; HT501128-4L, Sigma Aldrich, Overijse, Belgium) for 4 h at 4 °C, before paraffin embedding. Sections of 6 µm were deparaffinized and antigen retrieval was performed using citrate (pH 6.0) or Tris-EDTA (pH 9.0) buffer in Lab Vision PT Module (Thermo Fisher Scientific, Merelbeke, Belgium). After permeabilization with PBS/0.3% Triton-100X (3051.4, Carl Roth, Karlsruhe, Germany) for 5 min, sections were blocked with solution 1 (3% low-fat milk, 5% BSA, 0.3% Triton-100X in PBS) for 45 min, at RT. Primary antibodies (Table 1) diluted in solution 1 were added and incubated overnight, at 4 °C. The next day, slides were washed with PBS/0.1% Triton-100X and incubated with secondary antibodies diluted in solution 2 (10% BSA, 0.3% Triton-100X) at 37 °C, for 1 h, before DAB staining (ab64238, Abcam, UK, Cambridge). Pictures were selected after scanning on Pannoramic P250 Flash III (3DHistech, Budapest, Hungary). Edematous areas were measured on hematoxylin and eosin (H&E)-stained sections (measurements were performed on the whole pancreas area) using the CaseViewer software (3DHistech, Budapest, Hungary). The quantification of CD45, NF-κB(p65) and P-ERK^T202/Y204^ staining was performed on whole pancreas using the HALO software (Indica Labs, Albuquerque, NM, USA).

### 2.14. Statistical Analysis 

Data were presented as means ± standard error of the mean (SEM). Normality and equal variance were checked by applying a combination of four tests: Anderson–Darling test, D’Agostino–Pearson omnibus normality test, Shapiro–Wilk normality test and Kolmogorov–Smirnov normality test. Comparisons between two groups were performed using an unpaired Student’s t-test. Comparisons between three or more groups was performed using one-way analysis of variance (ANOVA). Significant effects or interactions were further analyzed by post hoc method. For all statistical analysis, the level of significance was set at *p* < 0.05. A statistical tendency was considered when *P*-values were 0.05 < *p* < 0.1. Analyses were performed using SigmaStat (version 3.1, Systat Software inc., San Jose, CA, USA) and GraphPad Prism (version 8, GraphPad Software, Inc., San Diego, CA, USA). For RNA-SEQ data, DESeq2 R package (Bioconductor), was used to determine significant differential gene expression using a model based on the negative binomial distribution. *P* values were corrected using the Benjamini and Hochberg’s approach for controlling the false discovery rate (FDR). Genes with a corrected *p* value < 0.05 were assigned as differentially expressed. * *p* < 0.05, ** *p* < 0.01, *** *p* < 0.001.

## 3. Results and Discussion 

### 3.1. Pancreatitis Induces a Greater Pro-Oxidant Response in Mice Bearing Oncogenic Kras^G12D^

To investigate the main pancreatitis-driven redox changes occurring in the presence of either wild-type *Kras* or mutated *Kras^G12D^* alleles, *WT* and *Ela-Kras^G12D^* mice were challenged with cerulein for 5 days to induce pancreatitis (inflammation of the pancreas). We first measured the oxidative damage marker 4-HNE. Interestingly, in total pancreas lysates, we found increased 4-HNE levels only in cerulein-treated *Ela-Kras^G12D^,* but not cerulein-treated *WT* mice (Figure 1A); this increase was also observed in the plasma of cerulein-treated *Ela-Kras^G12D^* mice compared to their untreated siblings (Appendix A), indicating that pancreatitis induces both local and systemic oxidative damages, selectively in *Ela-Kras^G12D^* mice. It is well-known that pancreatitis induces redox unbalance in *WT* rodents [9,10,11]. The absence of a significant increase in 4-HNE in our cerulein-treated *WT* mouse model is not synonymous with the absence of redox alterations; it probably means that a longer period and/or higher doses of cerulein treatment are needed to induce significant oxidative damage level. However, our data undoubtedly show that the oncogenic *Kras^G12D^* increases the extent of oxidative damages and accelerates the establishment of a pro-oxidant state in the pancreas. Since acinar cells are mainly affected by pancreatitis, we wanted to confirm that the redox changes observed in the total pancreas are also occurring in acinar cells. To do so, we used an ex vivo culture system that mimics the effect of pancreatitis on acinar cells. On day 3 (metaplastic acini), ROS levels were increased three-fold compared to day 1 (non-metaplastic acini) in acinar cultures from *Ela-Kras^G12D^* pancreata (Appendix A), indicating the presence of a pro-oxidant response in the whole pancreas and in acinar cells during pancreatic transformation. 

In addition, the expression of major redox-sensitive antioxidant and pro-oxidant enzymes was modulated by pancreatitis (Figure 1B and Appendix A). For example, the expression of peroxiredoxin-I (PRX-I), a major mammalian peroxidase, was similarly increased in cerulein-treated *Ela-Kras^G12D^* and *WT* mice compared to their untreated controls, respectively (Figure 1B); however, the expression of sulfiredoxin (SRX) and catalase (CAT) was selectively increased in cerulein-treated *Ela-Kras^G12D^* animals (Figure 1B). These data suggest enhanced antioxidant defense to buffer the pro-oxidant response induced by pancreatitis in the pancreas. We observed in FACS-sorted acinar cells, from *Ela-Kras^G12D^* pancreata, an increased expression of numerous antioxidant genes (Appendix A), including *Sulfiredoxin-1* (*Srxn1*), *thioredoxin-1* (*Txn1*) and *thioredoxin reductase 1* (*Txnrd1*), thus confirming that similar antioxidant defense is also taking place in acinar cells from cerulein-treated pancreata. Altogether, our data show that although pancreatitis promoted subtle redox alterations in *WT* mice, its combination with *Kras^G12D^* mutation induced the greatest level of pro-oxidant response and oxidative damages in the pancreas. 

### 3.2. Pancreatitis Modulates the Expression of Glutathione-Regulating Enzymes in Acinar Cells

After mapping the overall redox changes, we aimed to identify the most relevant redox mechanisms occurring in acinar cells during pancreatitis-driven tumorigenesis. We first analyzed the transcriptional landscape of FACS-sorted acinar cells from cerulein-treated and untreated *Ela-Kras^G12D^* mice and identified an upregulation in the “glutathione metabolism” pathway in response to pancreatitis (Figure 2A). This pathway was also upregulated, but to a lesser extent, in metaplastic acinar cells from cerulein-treated *WT* mice (Figure 2A). Interestingly, inflammation, cytokine production and pancreatic cancer pathways were all upregulated in the *Ela-Kras^G12D^* compared to the *WT* group (data not shown), suggesting that the inflammatory and pro-oxidant responses of pancreatitis are more severe in the former. 

Among other genes, we observed that the expression of *glutathione peroxidase* (*Gpx1*) was mostly increased at the mRNA level in FACS-sorted acinar cells from cerulein-treated *Ela-Kras^G12D^* mice compared to untreated counterparts (Figure 2B,C); glutathione peroxidase (GPx1) plays a critical role in the consumption of intracellular glutathione to buffer pro-oxidative responses. This result was also reproduced at the protein level in total pancreas lysates from cerulein-treated *Ela-Kras^G12D^* mice and in specific acinar cultures from the same genetic background (Figure 2D,E). Interestingly, the expression of GPx1 was higher in *Ela-Kras^G12D^* mice compared to their *WT* counterparts (Figure 2D), thus confirming the higher extent of redox changes occurring in the presence of *Kras^G12D^* mutation. We also observed an increased protein expression of GPx1 in primary human pancreatic cells cultured ex vivo under conditions mimicking inflammation (Figure 2F). As a proof of glutathione metabolism activation, we observed that short-term pancreatitis (1 day of treatment) decreased reduced glutathione (GSH) levels and increased the production of glutathione disulfide (GSSG) in the pancreas of cerulein-treated mice compared to healthy siblings (No cerulein) (Appendix A), highlighting the use of reduced glutathione (GSH) by GPx1 to buffer the pro-oxidant environment induced by pancreatitis [30]. These experiments validate the increase in GPx1 expression at the mRNA and protein levels and suggest that changes in glutathione utilization rates occur during pancreatitis. These experiments also reveal that a similar mechanism may also operate in inflamed human pancreata.

### 3.3. N-Acetylcysteine Reduces the Severity of Pancreatitis in Wild-Type and Ela-Kras^G12D^ Mice

As glutathione is depleted during pancreatitis, we questioned the usefulness of the thiol-containing compound, NAC, on pancreatic redox homeostasis and inflammation severity. Early studies have reported a positive antioxidant effect of NAC during pancreatitis in *WT* mice and human cells from patients with redox disorders [31,32,33,34,35,36]. However, the exact impact of NAC on pancreatic inflammation in a *Kras^G12D^* background is still not clearly elucidated. To induce acute pancreatitis, *WT* and *Ela-Kras^G12D^* mice received a 5-day cerulein regimen and were treated with NAC by intraperitoneal injection, as depicted in Figure 3. Acute pancreatitis increased GSSG levels and decreased GSH levels in *WT* and *Ela-Kras^G12D^* mice compared to untreated counterparts (no cerulein; Figure 4A and Figure 5A); these effects were more pronounced in *Ela-Kras^G12D^* animals, which is consistent with the higher level of oxidative damages observed in this background (Figure 1). NAC treatment increased the content of GSH and concomitantly decreased GSSG rates in the pancreata of cerulein-treated *WT* and *Ela-Kras^G12D^* mice (Figure 4A and Figure 5A); this is probably due to the direct scavenging of ROS by NAC, thereby sparing the intra-pancreatic GSH pool and/or to the ability of NAC to directly provide cysteine for GSH synthesis [37]. In both *WT* and *Ela-Kras^G12D^* groups, the total pancreas weight of NAC-treated mice was significantly reduced compared to that of cerulein-treated mice (Figure 4B and Figure 5B). Circulating amylase levels, as well as AST and ALT rates, increase during pancreatitis due to pancreas and liver damage, respectively [9]. NAC treatment significantly reduced the circulating levels of amylase, AST and ALT only in cerulein-treated *Ela-Kras^G12D^*, but not in *WT* mice; for the latter, a protective tendency was observed for circulating amylase levels without reaching a statistical significance (Figure 4C and Figure 5C). These measurements suggest that NAC reduces zymogen activation in acinar cells and relieves pancreatitis-related liver damage, especially in the *Ela-Kras^G12D^* background.

Histological analysis showed a better pancreas architecture in NAC-treated groups (Appendix A). Whole-tissue quantification of H&E-stained sections revealed that NAC treatment decreased edema by two folds in both cerulein-treated *WT* and *Ela-Kras^G12D^* mice (Figure 4D,E and Figure 5D,E). Similarly, collagen deposit, stained by Sirius Red, was also reduced by 1.1- and 1.4-fold in response to NAC treatment in cerulein-treated *WT* and *Ela-Kras^G12D^* mice, respectively (Figure 4D,E and Figure 5D,E). CD45 is a transmembrane receptor expressed on the majority of immune cells and used as a marker to measure tissular immune cell infiltration. Although NAC did not impact CD45-positive immune cell infiltration (Appendix A), it significantly reduced oxidative damage markers at the pancreatic (i.e., protein carbonyls) and systemic (i.e., 4-HNE) levels in both *WT* and *Ela-Kras^G12D^* backgrounds (Figure 4F, Figure 5F and Appendix A). Since immune cells are a major source of ROS production during inflammation [38], our results suggest that NAC is correcting redox imbalance by directly scavenging ROS generated by immune cells rather than affecting their infiltration in the pancreas. Accordingly, the activity of myeloperoxidase (MPO), a pro-oxidant enzyme majorly expressed by immune cells, was decreased particularly in *Ela-Kras^G12D^* pancreata suffering from pancreatitis (Figure 4G and Figure 5G). Our data indicate that NAC alleviates the hallmarks of pancreatitis and related oxidative damages, especially in the *Ela-Kras^G12D^* genotype.

### 3.4. N-Acetylcysteine Improves Pancreatitis-Related Inflammation in Ela-Kras^G12D^ Mice

Since *Ela-Kras^G12D^* mice exhibited a higher responsiveness to NAC treatment, we focused on this genetic background to better understand the impact of NAC on pancreatic inflammation during pancreatitis. We returned to the transcriptomic data of FACS-sorted acinar cells from cerulein-treated and untreated *Ela-Kras^G12D^* mice and found an important upregulation of TNF signaling in response to pancreatitis (Figure 6A). Since TNF-α plays a key role in the formation of edema in the lungs [39,40,41], we assume that it can exert similar functions in the pancreas. To test our hypothesis, using ELISA, we measured the TNF-α content in pancreas lysates from cerulein-treated *Ela-Kras^G12D^* mice supplemented with PBS or NAC. Interestingly, in the pancreas, edema levels correlated positively with TNF-α levels (Pearson, r = 0.93, *p* < 0.05) and NAC treatment reduced TNF-α rates in the pancreata of *Ela-Kras^G12D^* mice compared to vehicle-treated counterparts (Figure 6B). Acinar cells actively produce pro-inflammatory cytokines during pancreatitis and pancreas tumorigenesis [21,42]. To check the impact of NAC on acinar-specific TNF-α release, we isolated acinar cells from *Ela-Kras^G12D^* pancreata and cultured them ex vivo under conditions mimicking inflammation. Strikingly, NAC-treated acinar cells released two-fold less TNF-α in culture media compared to vehicle-treated ones (Figure 6C). Accordingly, the nuclear translocation of NF-κB(p65), a transcription factor involved in the regulation of cytokines expression [43], was also decreased by two-fold in acinar cells from NAC-treated *Ela-Kras^G12D^* pancreata (Figure 6D,E). In favor of a transcriptional control for TNF-α expression and release, we found that NAC treatment significantly reduced *TNF-α* mRNA expression in ex vivo cultured acinar cells from *Ela-Kras^G12D^* pancreata (Appendix A); the expression of other pro-oxidant and inflammatory genes, including *IL6*, *Ccl2* and *Nos2*, was downregulated in NAC-treated acinar cells compared to vehicle-treated cultures (Appendix A). Finally, the activation of ERK-MAPK, a critical pathway for pancreas carcinogenesis and inflammation, tended to decrease in acinar cells following NAC administration into *Ela-Kras^G12D^* mice (Figure 6D,E). However, we cannot rule out that NAC may also reduce the production of TNF-α from immune cells infiltrating the pancreas. Our results indicate that NAC reduces the severity of inflammation during the initiation of pancreas tumorigenesis, at least partly, by decreasing the production of TNF-α from acinar cells. 

This is the first study comparing the impact of NAC on pancreatitis in mice bearing either *wild-type* or mutated *Kras^G12D^* alleles. Mutated *Kras^G12D^* is an essential driver of pancreatic carcinogenesis [4,5]. NAC reproducibly reduced the severity of pancreatitis in both *wild-type* and *Kras^G12D^*-bearing mice, with a higher efficiency in the later; this better responsiveness is probably due to the higher extent of inflammation and oxidative damages induced by *Kras^G12D^*, which renders the protective effects of NAC more pronounced and visible. The use of NAC is already approved in the clinics and its prescription to patients suffering from pancreatitis should be safe, with few side-effects. Interestingly, our data suggest that NAC could also be effective in reducing the severity of inflammation during the early tumorigenesis process. However, one limitation of the study is that the main molecular mechanism through which NAC mediates its protective effect is still not clearly elucidated, because NAC impacts a multitude of pathways that renders this identification extremely challenging. Additionally, although thiol-containing compounds were shown to exert anticancer effects on cultured pancreatic cancer cells [44,45], NAC failed to reduce the development of acinar-to-ductal metaplasia and neoplastic lesions in our model of mice bearing *Kras^G12D^* mutation (data not shown), indicating that it could be used as an adjuvant treatment to relieve inflammation, but not as a therapy for pancreatic cancer.

## 4. Conclusions

In this paper, we demonstrate that acute pancreatitis induced systemic and pancreatic oxidative damages leading to the activation of an antioxidant defense mechanism in acinar cells, by modulating the expression of redox enzymes. A relevant defense mechanism mobilized by acinar cells was the use of intracellular GSH by GPx1 to neutralize ROS and reduce the oxidative stress response. Providing an extracellular source of thiols by administrating NAC to mice increased intracellular GSH pool in the pancreas. NAC also reduced oxidative damages, inflammation hallmarks and pancreatic edema; this was evident especially when pancreatitis and *Kras^G12D^* mutation were combined. Importantly, the resorption of pancreatic edema is clinically associated with better health outcomes [46]. Edema is a protein-rich fluid that may form plugs and calcify in pancreatic ducts, leading to pressure, duct damages and chronic pain. We therefore consider the use of an edema-eliminating compound such as NAC to be clinically relevant. In addition, NAC treatment seems to be safe without any significant side-effects. It is already prescribed to relieve pulmonary mucus in patients [47] and our study reveals that it could be used similarly to reduce pancreatic edema. However, we do not consider NAC as a complete treatment for pancreatitis or pancreatic cancer, but rather as an adjuvant therapy to alleviate the symptoms of patients suffering from severe pancreatitis or pancreas malignancies. Our study encourages future testing of treatments combining NAC with anti-inflammatory and anticancer drugs in the context of pancreatitis and pancreatic cancer, respectively. 

## Figures and Tables

**Figure 1 antioxidants-10-01107-f001:**
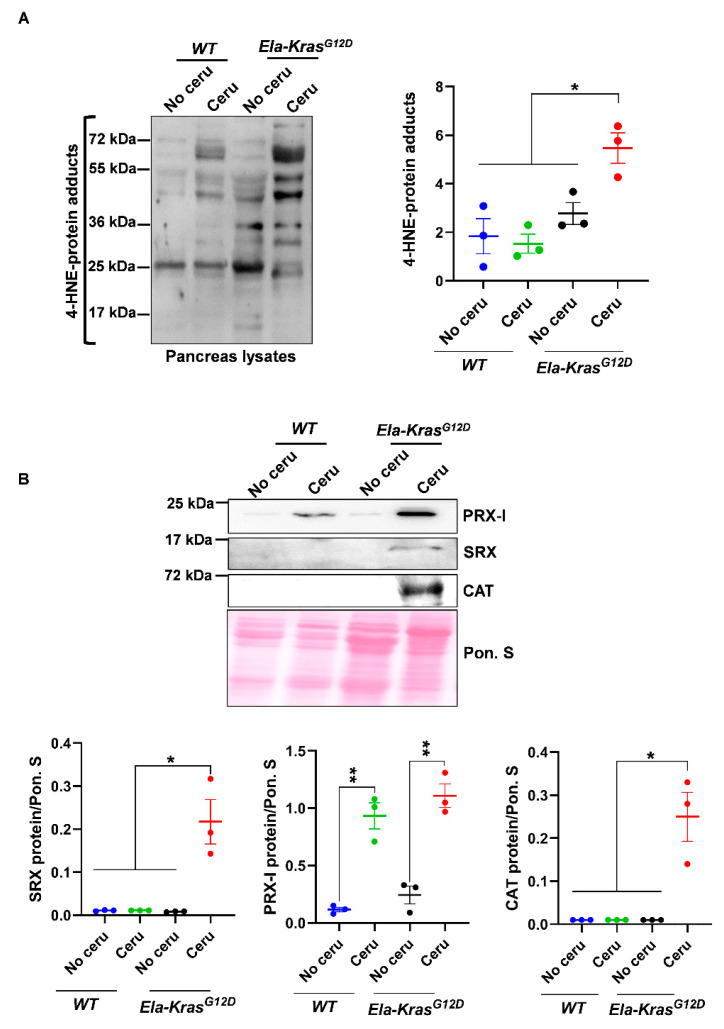
Pancreatitis induces an important pro-oxidant response particularly in *Ela-Kras^G12D^* mice. (**A**) Western blot for the oxidative damage marker, 4-HNE, on pancreas lysates of *wild-type* (*WT*) and *Ela-Kras^G12D^* mice treated with cerulein for 5 days (*n* = 3 for both *WT* and *Ela-Kras^G12D^*) or untreated (*n* = 3 for both *WT* and *Ela-Kras^G12D^*). The corresponding densitometry analysis is also available as a bar graph. (**B**) Western blot on pancreas lysates of *wild-type* (*WT*) and *Ela-Kras^G12D^* mice treated with cerulein for 5 days (*n* = 3 for both *WT* and *Ela-Kras^G12D^*) or untreated (*n* = 3 for both *WT* and *Ela-Kras^G12D^*). The analyzed enzymes are Peroxiredoxin-I (PRX-I), Sulfiredoxin (SRX) and Catalase (CAT). The corresponding densitometry analysis normalized to Ponceau S staining (Pon. S) is also available as a bar graph. Data are mean ± SEM. Statistical significance for all panels in this figure was tested by one-way ANOVA test and further interactions were analyzed by post hoc test (* *p* < 0.05; ** *p* < 0.01). All quantifications in this figure show biological replicates.

**Figure 2 antioxidants-10-01107-f002:**
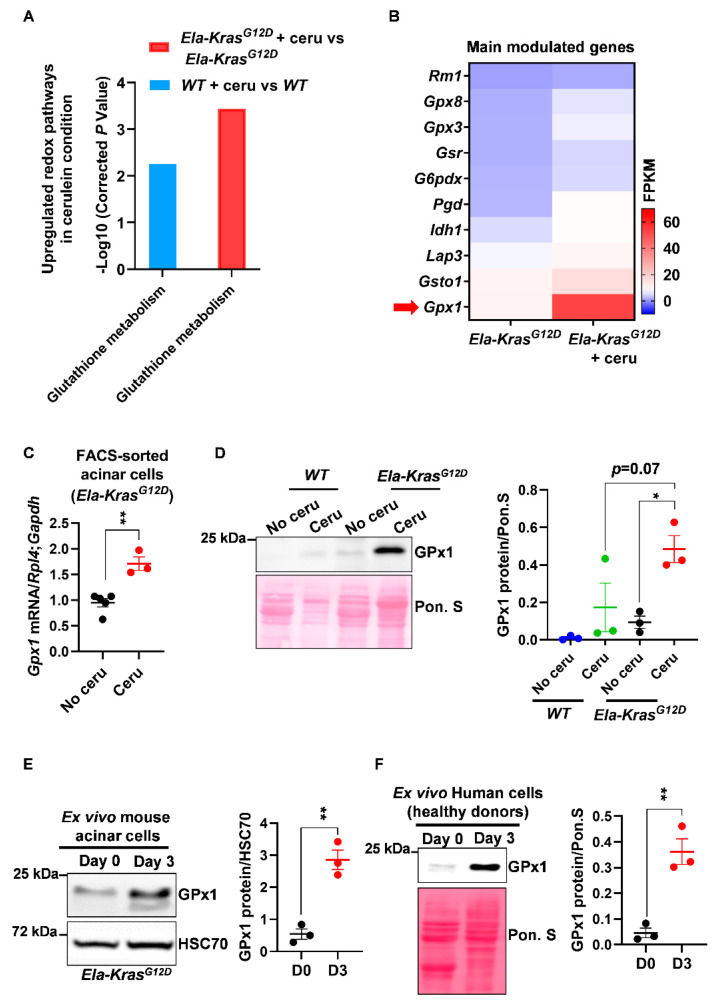
Pancreatitis modulates the expression of glutathione-regulating enzymes in mouse and human pancreatic cells. (**A**) Upregulated glutathione pathway identified by KEGG analysis in FACS-sorted acinar cells from *wild-type* (*WT)* and *Ela-Kras^G12D^* mice treated with cerulein for 3 days (*n* = 3 for both *WT* and *Ela-Kras^G12D^*) or untreated (*n* = 3 for both *WT* and *Ela-Kras^G12D^*). “Glutathione metabolism” is the official term used in the KEGG database; it means that the expression of a set of genes involved in glutathione oxidation and/or reduction is being changed. The corrected *P*-values were presented as “-Log10” so that when the differential expression is higher, the *P*-value on graph follows the same tendency. (**B**) Heatmap for the main genes modulated in the glutathione metabolism pathway in *Ela-Kras^G12D^* mice treated with cerulein for 3 days (*n* = 3) or untreated (*n* = 3). Red arrow shows one of the most expressed genes, *GPx1*. (**C**) Quantitative PCR for *glutathione peroxidase 1* (*Gpx1*) on FACS-sorted acinar cells from *Ela-Kras^G12D^* mice treated with cerulein for 3 days (*n* = 3) or untreated (*n* = 5). *Ribosomal protein L4* (*Rpl4*) and *glyceraldehyde-3-phosphate dehydrogenase* (*Gapdh*) were used as housekeeping genes. (**D**) Western blot for GPx1 on pancreas lysates of *wild-type* (*WT*) and *Ela-Kras^G12D^* mice treated with cerulein for 5 days (*n* = 3 for both *WT* and *Ela-Kras^G12D^*) or untreated (*n* = 3 for both *WT* and *Ela-Kras^G12D^*). The corresponding densitometry analysis normalized to Ponceau S (Pon. S) is also available as a bar graph. (**E**) Western blot for GPx1 in ex vivo cultured acinar cells on day 0 (normal) and 3 (metaplasia); these cells were isolated from *Ela-Kras^G12D^* pancreata and cultured under conditions mimicking inflammation (*n* = 3). The corresponding densitometry analysis normalized to HSC70 is also available as a bar graph. Three independent cultures from three different mice were tested. (**F**) Western blot for GPx1 in ex vivo cultured human acinar cells on day 0 (normal) and 3 (metaplasia); these cells were obtained from healthy heart-beating cadaveric donors and cultured under conditions mimicking inflammation (*n* = 3). Three independent cultures from three different donors were tested. Due to the scarcity of human material, we only assessed the expression of GPx1, which is the most relevant redox enzyme for this study. The corresponding densitometry analysis normalized to Ponceau S (Pon. S) is also available as a bar graph. Data are mean ± SEM. *P*-values for panel **A** were calculated using a Student’s t-test and corrected by the Benjamini and Hochberg method. Statistical significance for Panels **C**, **E** and **F** was tested by Student’s t-test (** *p* < 0.01). Statistical significance for panel **D** was tested by one-way ANOVA test and further interactions were analyzed by post hoc test (* *p* < 0.05). A statistical tendency was considered when *p*-values were 0.05 < *p* < 0.1. All quantifications in this figure show biological replicates.

**Figure 3 antioxidants-10-01107-f003:**
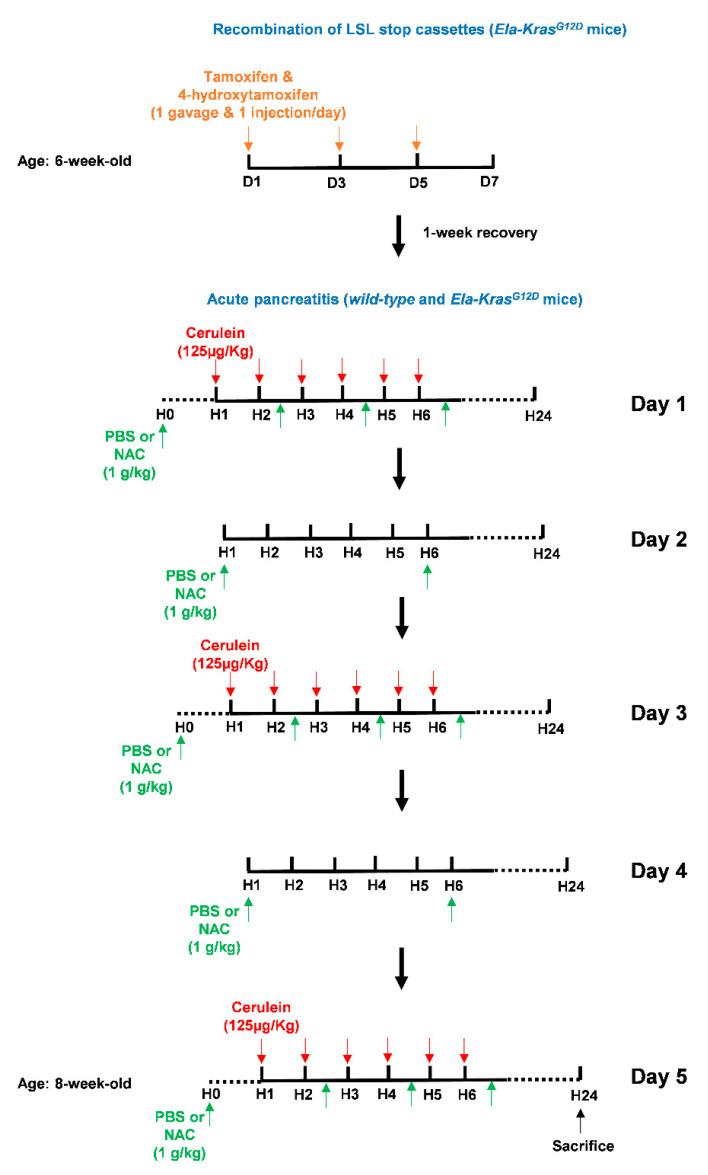
Illustration of the tamoxifen and cerulein regimens as well as NAC treatments. To initiate pancreas tumorigenesis, 6-week-old *Ela-Kras^G12D^* mice were first treated once a day with tamoxifen (gavage) and 4-hydroxytamoxifen (subcutaneous injection), on day (D) 1, 3 and 5 (orange arrows), to induce recombination of the LSL stop cassette and allow the expression of *Kras^G12D^* from its endogenous locus. After tamoxifen injections, mice recovered for 1 week. To induce acute pancreatitis, *wild-type* (*WT)* or tamoxifen-treated *Ela-Kras^G12D^* mice started a 5-day acute pancreatitis regimen. On day (D) 1, 3 and 5, mice received 6-daily intraperitoneal cerulein injections (red arrows) at a frequency of 1 injection/h (H). Cerulein diluted in a PBS volume of 100-150 µL was injected at a concentration of 125 µg/kg/injection. On day (D) 1, 3 and 5, mice received 4 intraperitoneal injections of NAC as indicated (green arrows). On day (D) 2 and 4, mice received 2 intraperitoneal injections of NAC as indicated (green arrows). Each NAC dose was 1 g/kg in a volume of 100–150 µL diluted in PBS. For vehicle-treated mice, intraperitoneal injections of 100–150 µL PBS were given instead of NAC. Mice were then sacrificed by cervical dislocation after 24 h.

**Figure 4 antioxidants-10-01107-f004:**
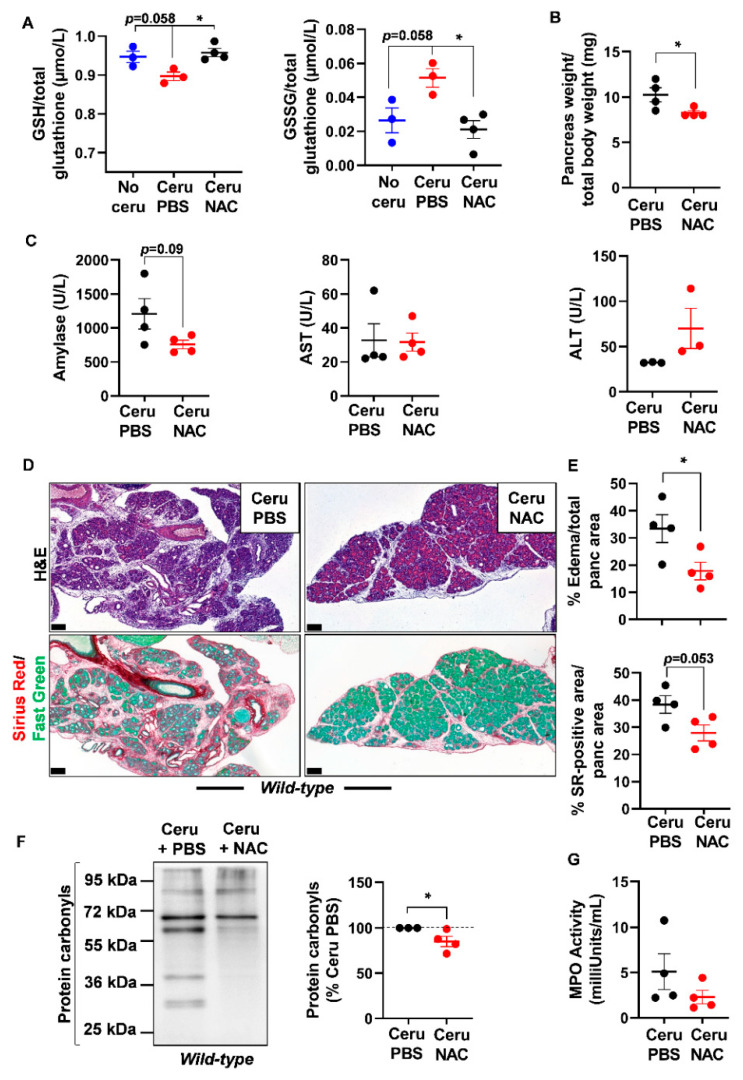
N-acetylcysteine treatment improves acute pancreatitis hallmarks in *wild-type* mice. (**A**) Reduced glutathione (GSH) and glutathione disulfide (GSSG) measurements in pancreas lysates of *wild-type* (*WT*) mice treated with cerulein and PBS for 5 days (Ceru PBS; *n* = 3), cerulein and NAC for 5 days (Ceru NAC; *n* = 4) or untreated (No ceru; *n* = 3). (**B**) Pancreas weight normalized to total body weight in *wild-type* (*WT*) mice treated with cerulein and PBS for 5 days (Ceru PBS; *n* = 4) or cerulein and NAC for 5 days (Ceru NAC; *n* = 4). (**C**) Measurement of plasma amylase*,* aspartate transaminase (AST) and alanine transaminase (ALT) levels in *wild-type* (*WT*) mice treated with cerulein and PBS for 5 days (Ceru PBS; *n* = 3–4) or cerulein and NAC for 5 days (Ceru NAC; *n* = 3–4). (**D**) Hematoxylin and eosin (H&E) staining and Sirius Red (SR) staining on the pancreata of *wild-type* (*WT*) mice treated with cerulein and PBS for 5 days (Ceru PBS; *n* = 4), or cerulein and NAC for 5 days (Ceru NAC; *n* = 4). Bars: 100 µm. (**E**) Whole-tissue quantifications for edema and SR-positive areas from panel **D**. (**F**) Western blot for the oxidative damage marker, protein carbonyls, on pancreas lysates of *wild-type* (*WT*) mice treated with cerulein and PBS for 5 days (Ceru PBS; *n* = 3) or cerulein and NAC for 5 days (Ceru NAC; *n* = 4). (**G**) Myeloperoxidase (MPO) activity on pancreas lysates of *wild-type* (*WT*) mice treated with cerulein and PBS for 5 days (Ceru PBS; *n* = 4) or cerulein and NAC for 5 days (Ceru NAC; *n* = 4). Data are mean ± SEM. Statistical significance for Panel **A** was tested by one-way ANOVA test and further interactions were analyzed by post hoc test (* *p* < 0.05). Statistical significance for panels **B**–**G** was tested by Student’s *t*-test (* *p* <0.05). Dixon’s Q test was applied to remove outlier samples, before conducting statistical analysis. A statistical tendency was considered when *p*-values were 0.05 < *p* < 0.1. All quantifications in this figure show biological replicates.

**Figure 5 antioxidants-10-01107-f005:**
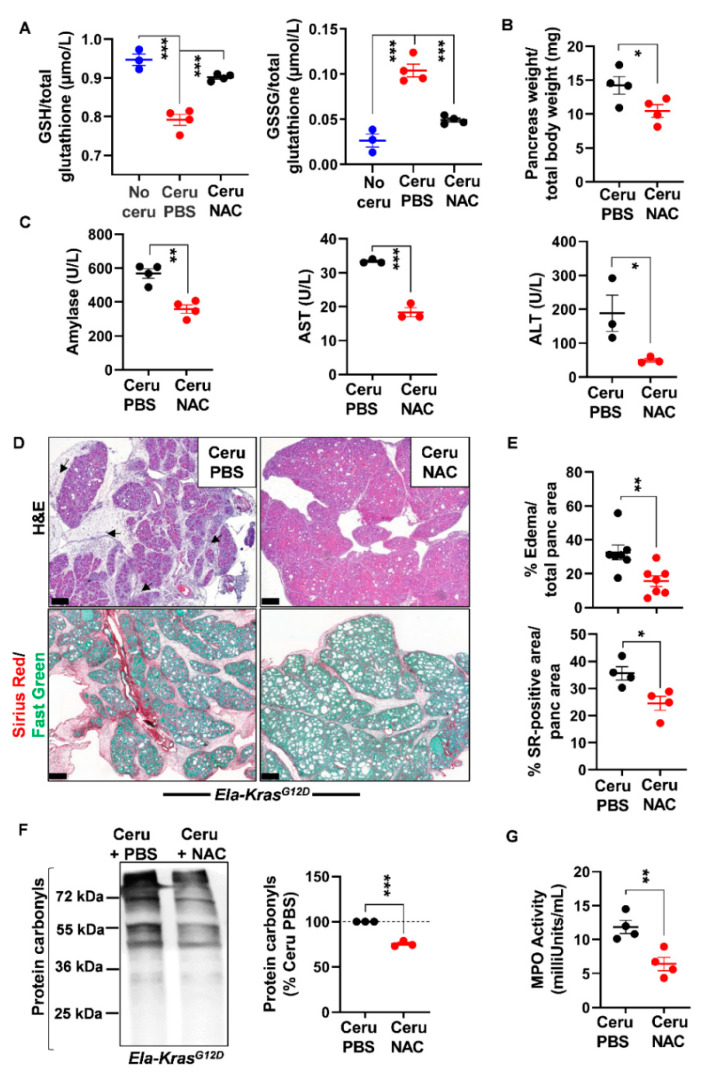
N-acetylcysteine treatment improves acute pancreatitis hallmarks in *Ela-Kras^G12D^* mice. (**A**) Reduced glutathione (GSH) and glutathione disulfide (GSSG) measurements in pancreas lysates of *Ela-Kras^G12D^* mice treated with cerulein and PBS for 5 days (Ceru PBS; *n* = 4), cerulein and NAC for 5 days (Ceru NAC; *n* = 4) or untreated (No ceru; *n* = 3). (**B**) Pancreas weight normalized to total body weight in *Ela-Kras^G12D^* mice treated with cerulein and PBS for 5 days (Ceru PBS; *n* = 4) or cerulein and NAC for 5 days (Ceru NAC; *n* = 4). (**C**) Measurement of plasma amylase*,* aspartate transaminase (AST) and alanine transaminase (ALT) levels in *Ela-Kras^G12D^* mice treated with cerulein and PBS for 5 days (Ceru PBS; *n* = 3–4) or cerulein and NAC for 5 days (Ceru NAC; *n* = 3–4). (**D**) Hematoxylin and eosin (H&E) staining and Sirius Red (SR) staining on the pancreata of *Ela-Kras^G12D^* mice treated with cerulein and PBS for 5 days (Ceru PBS; *n* = 4–7) or cerulein and NAC for 5 days (Ceru NAC; *n* = 4–7). Bars: 200 µm. Black arrows show edematous area. (**E**) Whole-tissue quantifications for edema- and SR-positive areas from panel **D**. (**F**) Western blot for the oxidative damage marker, protein carbonyls, on pancreas lysates of *Ela-Kras^G12D^* mice treated with cerulein and PBS for 5 days (Ceru PBS; *n* = 3) or cerulein and NAC for 5 days (Ceru NAC; *n* = 3). (**G**) Myeloperoxidase (MPO) activity on pancreas lysates of *Ela-Kras^G12D^* mice treated with cerulein and PBS for 5 days (Ceru PBS; *n* = 4) or cerulein and NAC for 5 days (Ceru NAC; *n* = 4). Data are mean ± SEM. Statistical significance for Panel **A** was tested by one-way ANOVA test and further interactions were analyzed by post hoc test (*** *p* < 0.001). Statistical significance for panels **B**–**G** was tested by Student’s *t*-test (* *p* < 0.05; ** *p* < 0.01, *** *p* < 0.001). Dixon’s Q test was applied to remove outlier samples, before conducting statistical analysis. All quantifications in this figure show biological replicates.

**Figure 6 antioxidants-10-01107-f006:**
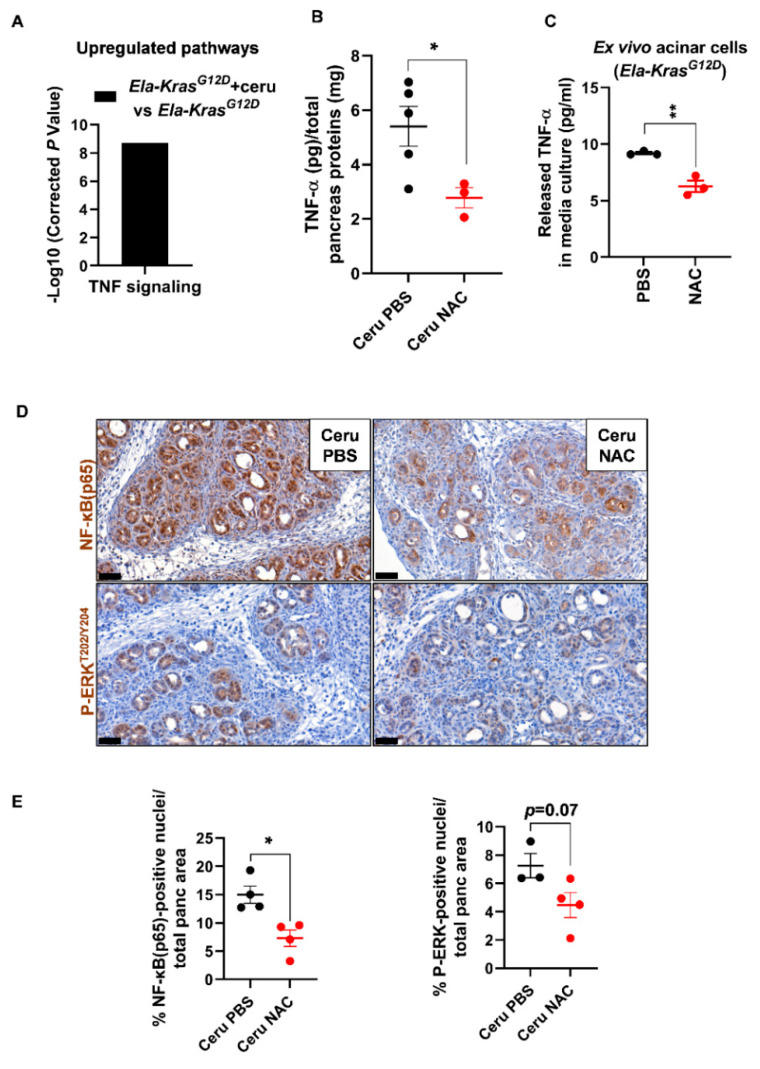
N-acetylcysteine reduces TNF-α production from acinar cells in *Ela-Kras^G12D^* mice. (**A**) TNF signaling identified as a major upregulated pathway by KEGG analysis in FACS-sorted acinar cells from *Ela-Kras^G12D^* mice treated with cerulein for 3 days (*n* = 3). (**B**) TNF-α protein measurements by ELISA, in pancreas lysates from *Ela-Kras^G12D^* mice treated with cerulein and PBS for 5 days (Ceru PBS; *n* = 5) or cerulein and NAC for 5 days (Ceru NAC; *n* = 3). Total pancreas protein concentration was determined by a Bradford assay and used for normalization of TNF-α protein levels. (**C**) TNF-α protein measurements by ELISA in the culture media of acinar cells isolated from *Ela-Kras^G12D^* mice and cultured under conditions mimicking inflammation. Acinar cells were treated with PBS or NAC (5 mM) from the onset of the culture until 3 days later (*n* = 3). Three independent cultures from three different mice were tested. (**D**) Immunohistochemistry for NF-κB(p65) and P-ERK^T202/Y204^ on the pancreata of *Ela-Kras^G12D^* mice treated with cerulein and PBS for 5 days (Ceru PBS; *n* = 3–4) or cerulein and NAC for 5 days (Ceru NAC; *n* = 4). Bars: 50 µm. (**E**) Whole-tissue quantifications for NF-κB(p65) and P-ERK^T202/Y204^ from panel **D**. Data are mean ± SEM. Statistical significance for all panels was tested by Student’s t-test (* *p* < 0.05; ** *p* < 0.01). Dixon’s Q test was applied to remove outlier samples, before conducting statistical analysis. A statistical tendency was considered when *p*-values were 0.05 < *p* < 0.1. All quantifications in this figure show biological replicates.

**Table 1 antioxidants-10-01107-t001:** Antibodies used in western blot (WB) and immunohistochemistry (IHC) experiments. CST, cell signaling technology; SCB, Santa-Cruz biotechnology.

Antibody	Reference	Dilution	Application
Superoxide dismutase 1 (SOD1)	37385T, CST, Leiden, The Netherlands	1/1000	WB
Catalase (CAT)	12980T, CST, Leiden, The Netherlands	1/1000	WB
Sulfiredoxin (SRX)	14273-1-AP, Proteintech, Leon-Rot, Germany	1/500	WB
Xanthine oxidase (XO)	Sc398548, SCB, Heidelberg, Germany	1/500	WB
Mitogenic oxidase 1 (MOX1)	Sc518023, SCB, Heidelberg, Germany	1/500	WB
Peroxiredoxin-I (PRX-I)	NBP1-82558, Bio-Techne, Minneapolis, MN, USA	1/1000	WB
Glutathione peroxidase 1 (GPx1)	AF3798-SP, Bio-Techne, Minneapolis, MN, USA	1/1000	WB
4-Hydroxynonenal (4-HNE)	ab46545, Abcam, Cambridge, UK	1/500	WB
Protein carbonyls	ab178020, Abcam, Cambridge, UK	1/5000	WB
Heat shock cognate (HSC70)	Sc7298, SCB, Heidelberg, Germany	1/2000	WB
Nuclear factor Kappa B(NF-κB(p65))	8242S, CST, Leiden, The Netherlands	1/100	IHC
Phospho-Extracellular signal-regulated kinase (P-ERK^T202/Y204^)	9101S, CST, Leiden, The Netherlands	1/100	IHC
CD45	ab10558, Abcam, Cambridge, UK	1/100	IHC

**Table 2 antioxidants-10-01107-t002:** PCR primers used in the present study.

Gene	Accession Number	Forward (5′-3′)	Reverse (5′-3′)	Species
*Glutathione peroxidase 1 (Gpx1)*	14775	AGTCCACCGTGTATGCCTTC	GTGTCCGAACTGATTGCACG	*Mus musculus*
*Superoxide dismutase 1 (Sod1)*	20655	GGAACCATCCACTTCGAGCA	CTGCACTGGTACAGCCTTGT	*Mus musculus*
*Catalase (Cat)*	12359	CTCGCAGAGACCTGATGTCC	TGTGGAGAATCGAACGGCAA	*Mus musculus*
*Thioredoxin 1 (Txn1)*	22166	AAGCTTGTCGTGGTGGACTT	AACTCCCCCACCTTTTGACC	*Mus musculus*
*Thioredoxin-interacting protein (Txnip)*	56338	CCTAGTGATTGGCAGCAGGT	GAGAGTCGTCCACATCGTCC	*Mus musculus*
*Thioredoxin reductase 1 (Txnrd1)*	50493	AGAGCTGGTGGTTTCACCTTC	TTTTTGTTCGGCTTCAGGGCT	*Mus musculus*
*Sulfiredoxin (Srxn1)*	76650	GTACCAATCGCCGTGCTCAT	CTCACGAGCTTGGCAGGAAT	*Mus musculus*
*Tumor necrosis factor-α (TNF-α)*	21926	GTGACAAGCCTGTAGCCCAC	ACAAGGTACAACCCATCGGC	*Mus musculus*
*Interleukine 6 (IL6)*	16193	TGTTCTCTGGGAAATCGTGGA	AGCATTGGAAATTGGGGTAGGA	*Mus musculus*
*Interleukine 1-α (IL1-α)*	16175	CGCTTGAGTCGGCAAAGAAA	CTGATACTGTCACCCGGCTC	*Mus musculus*
*Interleukine 1-β (IL1-β)*	16176	GCCACCTTTTGACAGTGATGAG	AAGGTCCACGGGAAAGACAC	*Mus musculus*
*Nitric oxide synthase 2 (Nos2)*	18126	TGAAACTTCTCAGCCACCTTGG	AGAGAAACTTCCAGGGGCAAG	*Mus musculus*
*C-C motif chemokine ligand 2 (Ccl2)*	20296	CTGTGCTGACCCCAAGAAGG	AAGACCTTAGGGCAGATGCAG	*Mus musculus*
*Amylase (Amy2a5)*	109959	GTGGTCAATGGTCAGCCTTT	TTGCCATCGACCTTATCTCC	*Mus musculus*
*Chymotrypsin (Ctrc)*	76701	GGATGACACTTGGAGGCACA	CGATGTCGTTCCACAGCAGA	*Mus musculus*
*Ribosomal protein L4 (Rpl4)*	67891	CGCAACATCCCTGGTATTACT	TGTGCATGGGCAGGTTATAGT	*Mus musculus*
*Glyceraldehyde 3-phosphate dehydrogenase* (*Gapdh*)	14433	GGTCCTCAGTGTAGCCCAAG	AATGTGTCCGTCGTGGATCT	*Mus musculus*

## Data Availability

RNA-SEQ data can be downloaded from the gene expression omnibus (GEO database using the following accession numbers: GSE163254 and GSE163263.

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
