# Peer review of "N-Acetylcysteine Reduces the Pro-Oxidant and Inflammatory Responses during Pancreatitis and Pancreas Tumorigenesis"

_antioxidants, 2021, doi:10.3390/antiox10071107_

Round 1

Reviewer 1 Report

file attached

Author Response

Dear reviewer, 

We thank you for your suggestions that improved the quality of our manuscript. 

Please find our point-by-point reply in the enclosed PDF file.

Sincerely,

Reviewer 2 Report

The rationale for the study is unclear as the introduction is a bit confusing, comprising several pieces of apparently unlinked information. A more integrated appraisal of the relevant literature would be appropriate to provide the context for the study.

Immunoblots (representative of how many experiments?) must be cropped in a way that retains information about antigen size and antibody specificity. The cropped images must retain sufficient area around the band(s) of interest, ideally including the positions of at least one molecular weight marker above and below the band(s). Moreover, to ensure reproducibility by other laboratories, the manufacturer name and catalog number(s) of all the antibodies used should be clearly specified in the methods section.

All legends should include specific "n" for each (and every) treatment group and a description of the statistics used for each experiment. Actually, the Authors must state numbers of independent samples (biological replicates) and replicate samples (technical replicates) analyzed, state clearly whether figures show technical or biological replicates, and report how many times each experiment was repeated. Define whether replicates are technical replicates or independent experimental replicates.

A dimensional bar is missing in figure 3B.

Statistics:

-how was normal distribution verified?

-ANOVA should be confirmed by post hoc analysis

The manuscript is mainly descriptive and focused on its (not fully supported) conclusions, not adequately acknowledging the limitations of the study. The strengths and limitations of the study should be deeply addressed, taking into account sources of potential bias or imprecision: Discuss both direction and magnitude of any potential bias.

It is advisable to the Authors to incorporate a pictorial or cartoon representation of the main results of the study to increase the overall impact of the manuscript.

Author Response

(The authors gave the same response as above.)

Reviewer 3 Report

This manuscript provides nice results about the redox changes associated with early pancreatic tumorigenesis after induction of acute pancreatitis. Although the results are clear, the study requires incorporating new controls and revising some unconvincing measures.

  1. Cerulein-induced acute pancreatitis causes strong redox modifications. This fact is well known in the field of acute pancreatitis. Therefore, the authors should clarify whether the redox modifications studied are increased in Ela-KrasG12D mice with acute pancreatitis compared to WT mice with pancreatitis. To do this, the authors must incorporate WT sham mice and WT + cerulein mice into their measurements.
  2. The 125 ug / kg dose of cerulein is too high to induce acute pancreatitis. In this model, doses of 50 ug / kg are normally used. The authors must justify this variation of the model.
  3. The use of NAC in the treatment of acute pancreatitis is a widely explored field. In the data shown in Figure 3 it is very surprising that acute pancreatitis does not cause depletion of glutathione levels. With 6 injections of cerulein and with the dose used in this study, pancreatic glutathione levels should decrease significantly. I recommend avoiding spectrophotometric techniques for determining glutathione levels and, of course, they should not be used to determine GSSG levels. Mass spectrometry must be used for these determinations. Furthermore, the levels of GSH, GSSG and GSH / GSSG should also be determined in the KrasG12D and KrasG12D + cerulein mice and compared with their WT counterparts.
  4. The authors state that their results suggest that NAC modulates ROS generated by immune cells. This fact should be clarified, for example, by measuring pancreatic MPO levels.
  5. The protective effect of NAC in acute pancreatitis is well documented. What about the effect of NAC on tumorogenesis in KrasG12D mice? The manuscript lacks measures that relate the effect of NAC on the transformation of pancreatic cells in KrasG12D + cerulein mice. If there is an effect, the authors should clarify whether it is due to antioxidant or anti-inflammatory protection mediated by NAC.

Author Response

(The authors gave the same response as above.)

Round 2

Reviewer 2 Report

The Authors did a great job in revising their manuscript.

The following reports should be mentioned:

PMID: 34067020 

PMID: 33672594 

PMID: 32322336

PMID: 30584464

PMID: 29636531 

PMID: 28928397 

PMID: 28370189

PMID: 28025489

Author Response

Dear Editors, 

We have addressed the last comments from reviewer#2. We have added the requested references in the text. These new changes are highlighted in yellow. 

Sincerely yours,

Patrick Jacquemin and Mohamad Assi

Reviewer 3 Report

The authors have answered most of my concerns. The manuscript is ready to be published. 

Author Response

(The authors gave the same response as above.)
